# The Andrological Landscape in the Twenty-First Century: Making Sense of the Advances in Male Infertility Management for the Busy Clinicians

**DOI:** 10.3390/ijerph21091222

**Published:** 2024-09-17

**Authors:** Ahmad Motawi, Andrea Crafa, Taha Hamoda, Rupin Shah, Ashok Agarwal

**Affiliations:** 1Department of Andrology, Sexual Medicine and STIs, Faculty of Medicine, Cairo University, Cairo 11956, Egypt; amotawi@kasralainy.edu.eg; 2Global Andrology Forum (GAF), Moreland Hills, OH 44022, USA; crafa.andrea@outlook.it (A.C.); tahaaboalmagd@yahoo.com (T.H.); rupinurvashishah@gmail.com (R.S.); 3Department of Clinical and Experimental Medicine, University of Catania, 95123 Catania, Italy; 4Department of Urology, King Abdulaziz University, Jeddah P.O. Box 80215, Saudi Arabia; 5Department of Urology, Faculty of Medicine, Minia University, Minia 2431436, Egypt; 6Department of Urology, Lilavati Hospital and Research Center, Mumbai 400050, India

**Keywords:** male infertility, diagnosis of male infertility, treatment of male infertility, varicocele, sperm DNA fragmentation, WHO Manual, omics, artificial intelligence, home semen analysis

## Abstract

Male infertility represents a significant global problem due to its essential health, social, and economic implications. It is unsurprising that scientific research is very active in this area and that advances in the diagnostic and therapeutic fields are notable. This review presents the main diagnostic advances in male infertility, starting from the changes made in the latest WHO Manual of semen analysis and discussing the more molecular aspects inherent to “omics”. Furthermore, the usefulness of artificial intelligence in male infertility diagnostics and the latest advances in varicocele diagnosis will be discussed. In particular, the diagnostic path of male infertility is increasingly moving towards a personalized approach to the search for the specific biomarkers of infertility and the prediction of treatment response. The treatment of male infertility remains empirical in many regards, but despite that, advances have been made to help formulate evidence-based recommendations. Varicocele, the most common correctable cause of male infertility, has been explored for expanded indications for repair. The following expanded indications were discussed: elevated sperm DNA fragmentation, hypogonadism, orchalgia, and the role of varicocele repair in non-obstructive azoospermia. Moving forward with the available data, we discussed the stepwise approach to surgical sperm retrieval techniques and the current measures that have been investigated for optimizing such patients before testicular sperm extraction. Finally, the key points and expert recommendations regarding the best practice for diagnosing and treating men with infertility were summarized to conclude this review.

## 1. Introduction

Infertility is a complex and multifaceted phenomenon that affects millions of people worldwide and has significant implications for the health and well-being of the affected individuals and couples. Despite that, there are still many fertility research gaps and practice blind spots. For instance, despite its long-standing use as the cornerstone routine test for male infertility, semen analysis has a limited ability to distinguish between men who suffer from infertility and men who do not [1]. Sperm DNA fragmentation (SDF), reactive oxygen species (ROS) measurements, and leukocyte detection tests are recent additions to the diagnostic armamentarium. Additionally, the intricate nature of the molecular interactions between oocyte and sperm that lead to fertilization remains unclear, despite the development of additional sperm tests, the zona penetration assay, the vitality and hypo-osmotic swelling test, and the mitochondrial activity index test [2]. Therefore, it is crucial to explore and address such existing gaps and propose future research directions to enhance knowledge and promote the scientific advancement of fertility management [3]. Future fertility research should target the development of new diagnostic tests that can identify sperm’s molecular and genetic defects so that personalized, targeted therapy can be offered.

The rapid development of medical informatics, machine learning, and artificial intelligence (AI), along with the demand for personalized medicine, have created new opportunities to contribute to modern healthcare. AI can help improve the diagnosis, treatment, prevention, and management of various health conditions and challenges. AI-based medical applications aim to deliver patient-centered precision medicine tailored to each patient’s needs and preferences and can help physicians in various clinical scenarios, such as inpatient, outpatient, and surgical care. Nevertheless, applying AI in andrology is still relatively new, and studies have only recently started to explore its potential [4,5].

Our review aims to identify gaps in current approaches, clarify the advancements needed in male infertility and how to utilize emerging scientific advances, and provide insights into what clinicians can offer to andrological patients in the near future.

## 2. Advances in Male Infertility Diagnostics

### 2.1. The Clinical Relevance and Implications of Changes in the sixth edition of the WHO Manual of Semen Analysis

One of the significant updates in the field of male infertility diagnostics that has characterized the last few years was the publication of the latest sixth edition of the WHO Manual of semen analysis [6]. Like every newer edition, the latest manual attempts to go beyond the limits of the previous edition. Among the main limitations of the previous manual was the cut-off of the sperm parameters that were established to distinguish fertile from patients with infertility [7]. Notably, these cut-off values were established in a small population that needed to be more genuinely representative of the world population. Men from areas with high population densities, such as Asia, the Middle East, Latin America, and Africa, were not represented [7].

Additionally, the previous manual recognizes that the use of a lower threshold (corresponding to the 5th percentile) for the diagnosis of infertility represents a limitation in the evaluation of males with infertility [8]. Moreover, for each patient, only one semen collection was considered, with a possible bias being determined by the wide intra-individual variability of the semen analysis [7]. Finally, the previous edition of the manual did not contemplate the use of any second-level examination (e.g., the evaluation of sperm DNA fragmentation), which helps characterize patients with idiopathic or unexplained infertility [7]. The sixth WHO Manual partly overcomes these limitations because it analyzes a larger and more geographically diverse population, including men from regions such as Asia, Africa, and Southern Europe [8]. It also introduces the evaluation of sperm DNA fragmentation (SDF) and the evaluation of genetic anomalies using the Fluorescent in situ hybridization (FISH) method in the extended examination to be performed in specific categories of patients [8]. The introduction of these tests will undoubtedly achieve a significant improvement in the identification of male infertility. However, even today, there are still significant limitations regarding the adequate standardization of these tests [9]. However, some limitations relating to the latest manual should be considered, such as the reintroduction of the distinction between slowly progressive and rapidly progressive motility, although no new studies clearly establish this distinction’s superiority on fertility outcomes [8,10]. Furthermore, the absence of clear reference limits, if it does not represent a problem for expert clinicians in the management of male infertility, could represent a barrier for clinicians who are less familiar with this condition, compromising their ability to choose whether to direct patients toward specialized management [8,11]. Table 1 summarizes the advantages and limitations of the latest edition of the WHO Manual of Semen Analysis.

### 2.2. Sperm DNA Fragmentation—Rationale, Testing Techniques, and Clinical Applications

As mentioned above, one of the main additions to the latest edition of the WHO Manual on sperm analysis is the introduction of SDF evaluation for use in clinical practice in specific categories of patients. This advancement of SDF examination from a test to be used for research purposes to a test to be used in clinical practice is a consequence of many studies published in the last decade about its usefulness in explaining some cases of idiopathic male infertility [12,13]. In addition, meta-analytic evidence would seem to suggest that a higher SDF rate is correlated with rising abortion rates, reduced embryo quality, and reduced clinical pregnancy rates in couples undergoing assisted reproductive technologies (ART). However, this evidence is weakened by the high heterogeneity of the included studies [14]. Another meta-analysis that included 13 prospective studies that assessed the difference in SDF between fertile couples and those with a history of recurrent pregnancy loss (RPL) showed substantially greater levels of SDF in male partners of couples with a history of RPL [15]. Despite this evidence, to date, there is still considerable controversy about for which patients and when to request an SDF analysis. A recent Global Andrology Forum (GAF) survey involving many clinicians from several different countries explored the clinical situations for which physicians order SDF testing, showing no uniformity of thought among them. This could be partly related to the absence of robust recommendations from professional societies’ guidelines concerning SDF testing as part of the investigation of men with infertility [16]. Another area for improvement is that, to date, there is no preferred method for measuring SDF, and the current recommendation is for each laboratory to establish and validate its threshold value [17]. In brief, the primary methods for SDF assessment are the terminal deoxynucleotidyl transferase-mediated deoxyuridine triphosphate nick end labeling (TUNEL) assay, the sperm chromatin structure assay (SCSA), the sperm chromatin dispersion (SCD) assay, and the Comet assay. Each of these methods has advantages and disadvantages. For example, TUNEL has high sensitivity and minimal interrater variability, making it extremely reliable; however, it is expensive and requires experienced personnel. Others, such as the Comet assay and the SCD, are less expensive and, in the case of the Comet assay, allow the evaluation even of samples with very low cell counts; however, they are characterized by high inter-observer variability, thus resulting in their reduced reliability [18]. Since, to date, none of the tests is considered superior to the other, the SDF assessment method should take into consideration the availability of resources, personnel, and the lab’s testing complexity. Moreover, given the close association between SDF and abstinence interval, it should be evaluated with an abstinence period of fewer than five days [17].

To date, the main indications for which an SDF analysis should be requested are unexplained and idiopathic infertility, recurrent pregnancy loss (RPL), clinical varicocele, and the patients’ exposure to lifestyle or environmental risk factors (advanced age, smoking tobacco, drugs and alcohol abuse, obesity, diabetes, ionizing radiation, environmental toxins, heat stressors, cryptorchidism, systemic inflammation, genital infection, or cancer). Moreover, its testing may be recommended either before or after the failure of ART and may be considered in the cryopreservation of sperm [16].

### 2.3. The Use of AI in Semen Analyses: Diagnostic Implication in Defining Treatment Direction

Advances in male infertility diagnostic research have also seen the involvement of artificial intelligence (AI), as in other fields of medicine (Figure 1). In this regard, the first prominent use of AI is in computer-aided sperm analyzer (CASA) systems. These are still categorized by the latest WHO Manual as “advanced examinations” for sperm evaluation, as the gold standard still appears to be a microscopic analysis performed by experienced and qualified personnel [6]. Using these systems allows a more objective and reproducible evaluation of the seminal fluid examination and a better assessment of the sperm’s kinetic parameters [19]. However, several limitations, such as the lack of standardization and the poor accuracy in assessing the sperm morphology and concentration in samples with high viscosity, severe oligozoospermia, or in the presence of many round cells, debris, and agglutination, prevent their routine use [20]. More recently, an integrated system of autofocus optical technology, AI algorithms, and electronic engineering mechanical systems have resulted in compact, rapid, observer-independent, and highly accurate CASA systems. A recent study demonstrated, with one such system, a good reliability and accuracy after comparison with a manually performed semen examination [21]. This suggests that the increasing enhancement of the algorithms integrated into these systems will increase their performance, improving the standardization, speed of execution, and objectivity of sperm analyses [21].

The use of AI in sperm analyses has another exciting application in the selection of sperm for use in assisted reproductive techniques (ARTs). For example, a study combining the use of high-resolution images of sperm morphology with a deep-convolutional neural network showed that the dry mass content ratios between the head, mid-piece, and tail of the spermatozoon could provide a prediction of successful rates for zygote cleavage and embryonic blastocyst formation [22]. Another study has highlighted how, through the training of machine learning algorithms with sets of sperm images, it is possible to identify spermatozoa with SDF with good reliability, which correlates with the ART procedure’s failure [23]. Therefore, applying AI to images resulting from sperm analyses could provide embryologists with a useful tool to create time-efficient, standardized, and reliable sperm selection processes, improving embryo and pregnancy outcomes [24].

### 2.4. Home Semen Analysis: Overview, Comparison, and Limitations

Although the manual microscopic evaluation of sperm parameters is the gold standard of male infertility diagnostics, many men are hesitant to undergo this test because it is experienced as invasive or as a social stigma. In this context, the use of at-home semen tests (Table 2), could be a convenient, low-cost means of initial fertility screening to decide whether or not to proceed with further investigations [25]. It has also been observed that the use of these methods improves the compliance of vasectomy patients to postoperative semen examination, thus allowing the evaluation of surgery success [26]. Furthermore, it has been observed that early home sperm testing for couples beginning to attempt to conceive has a favorable cost–effect relationship, since it can significantly reduce the time to male infertility evaluation and treatment, with potential benefits for the overall health outcomes for couples desiring pregnancy [27].

However, several limitations must be considered in using these tests, such as the lack of the possibility to perform a real quantification of sperm parameters. Only a qualitative evaluation is allowed to establish whether or not to carry out further investigations [25]. Furthermore, only one or a few parameters can be evaluated at a time, with the risk of false negatives due to the lack of the evaluation of parameters that may be fundamental for fertility [25]. Finally, the risk of false results, with consequent anxiety for the patient and increased costs for unnecessary investigations, cannot be ruled out [27].

### 2.5. Prospect of “Omics” in Male Infertility

The study of “omics” also represents a notable prospect of advancement in the diagnostics of male infertility. It has advanced rapidly in recent decades and includes genomics, transcriptomics, proteomics, epigenomics, and metabolomics [1]. While the traditional view regarded the spermatozoon as merely bearing the paternal DNA, current evidence suggests that spermatozoa carry many transcripts and proteins involved in several processes related to oocyte fertilization and embryo growth and development [28].

Not only do the transcripts and proteins carried by the spermatozoa play a role in fertility, but those present in the seminal plasma also seem to have a fundamental role in regulating the expression of the transcripts and proteins in the spermatozoa [29]. Thus, some miRNAs expressed in seminal plasma would appear to be predictive of the possibility of sperm recovery in TESE procedures in patients with non-obstructive azoospermia (NOA). For instance, proteins such as ECM1, TEX101, and LGALS3BP have been seen to help predict TESE outcomes in patients with NOA. In addition, the differential expression of ECM1 was also seen to be related to the success rate of ART [29]. Further, some miRNAs belonging to the miR-34, miR-122, and miR-509 families are down-regulated in patients with NOA and oligozoospermia [29]. Similarly, several seminal plasma proteins have been correlated with fertility. For example, a reduced expression of DJ-1, a protein involved in counteracting oxidative stress, is present in patients with asthenozoospermia [30].

Regarding epigenetics, the differential methylation of several imprinted genes has been correlated with alterations in fertility. A recent meta-analysis of 11 studies showed that the methylation of the H19 gene is reduced in patients with oligozoospermia and in patients with a history of RPL [31]. Similarly, the methylation of the mesoderm-specific transcription (MEST) gene is increased in patients with abnormal sperm parameters compared with healthy controls [32]. Another meta-analysis reached similar conclusions, showing that the Small Nuclear Ribonucleoprotein Polypeptide N (SNRPN) gene also exhibits significantly higher methylation in patients with infertility than in fertile ones [33]. Finally, the altered methylation of the Gene trap locus 2 (GTL2) gene has been associated with poorer ART outcomes [34].

Metabolomics also deserve a mention. Indeed, the substrates, products, and by-products of sperm anabolic and catabolic reactions could be used as indicators of the metabolic state of sperm. In particular, the high diagnostic value of ROS, total antioxidant capacity (TAC), oxidation-reduction potential (ORP), metabolite profiles, and individual lipids in sperm and seminal plasma for male sperm quality and fertility disorders has been shown [35]. Moreover, for outcome prediction, the use of 1H-NMR or MALDI-TOF MS methods for quantifying metabolite profiles in seminal plasma have a better performance than evaluating individual metabolites [35].

### 2.6. Genetic Tests in Men with Infertility—The Role and Status of Genome and Exome Sequencing

Genetics represents a hot topic in the diagnosis of male infertility. In fact, it is estimated that approximately 15% of the causes of infertility can be attributed to a genetic factor. Furthermore, numerous cases of idiopathic infertility seem to find their answer in a genetic alteration [36]. Given the close association between the two conditions, it is therefore not surprising that, to date, the main research topic associated with the study of male infertility is represented by genetics [37]. 

The methods of evaluating genetic alterations have also evolved with an increasingly in-depth study of gene sequences over the years. In fact, in addition to the classic karyotype examination, the search for the microdeletions of the AZF region, and the search for mutations in the CFTR genes or the central genes associated with hypogonadotropic hypogonadism, new methods have been introduced, which have allowed the identification of new targets that are responsible for the alterations in spermatogenesis [36]. For example, a genome-wide study using a single nucleotide polymorphism (SNP) array allowed the identification of two genes DYP19L and SPATA16 involved in the pathogenesis of globozoospermia [38,39]. 

The possibility of carrying out whole-exosome sequencing through next-generation sequencing (NGS) methods has allowed us to significantly improve our knowledge of the genetic causes of spermatogenetic failure [40]. A more recent systematic review published in 2021 identified 120 genes that were moderately, strongly, or definitively linked to 104 infertility phenotypes: a 33% increase compared to a previous analysis in 2019. This highlights how our understanding of the genetic nature of male infertility is growing rapidly [41].

The identification of more genes can partly explain idiopathic male infertility. Thus, a study of 25 patients with idiopathic oligozoospermia or NOA who underwent an NGS analysis of a panel of 15 genes found the presence of rare variants in the nuclear receptor subfamily 5, group A, member 1 (NR5A1) and testis-expressed 11 (TEX11) genes with a pathogenic role in 12.0% patients. Moreover, the study found seventeen other variants, of which, eleven were considered likely pathogenic and deserve functional or segregational studies. Among the genes most frequently mutated were MEIOB, USP9Y, KLHL10, NR5A1, and SOHLH1 [42]. Another study of 80 patients with NOA, no karyotype alterations, and no Yq AZF microdeletions analyzed the presence of mutations in testis-expressed TEX11, NR5A1, and double sex- and MAB3-related transcription factor 1 (DMRT1) genes, finding likely pathogenic mutations in four patients [40]. Therefore, NGS would allow the creation of patient-specific panels, improving the diagnosis of male infertility and thus reducing the rate of patients with idiopathic infertility diagnosis [42].

### 2.7. Advances in Varicocele Diagnosis

A quite commonly investigated topic in association with male infertility is represented by varicocele [37]. The reason why this topic still represents a source of considerable discussion is that despite varicocele being linked to male infertility, not all men with varicocele have infertility [43]. Additionally, although the beneficial effects of varicocele repair on sperm parameters have been demonstrated by various meta-analytic studies [44,45], to date, it is still controversial which parameters best predict the response to the repair of the varicocele [43]. In this sense, in recent years, efforts have increased to further our understanding of the predictive parameters of response, and the search for new diagnostic parameters has increased. 

The study of some ultrasound parameters has improved the predictive performance of the varicocele treatment. For example, the study of the testicular microcirculation in patients with varicocele has shown that patients with varicocele may present an increase in the pulsatility and resistive indices of the subcapsular branches of the testicular arteries and, in turn, these indices negatively correlate with motility and sperm count [46,47,48]. Another parameter that can be assessed by ultrasound examination is testicular stiffness, which can be assessed through shear wave elastosonography (SWE). Patients with varicocele whose SWE shows greater testicular stiffness have worse sperm parameters. Moreover, the extent of the reduction of this stiffness after surgery correlates with the improvement of sperm parameters. Therefore, this method would appear to be predictive of an improvement in sperm quality after varicocelectomy, proving to be useful for decision making with regard to the treatment of patients with varicocele [49]. However, although promising, the resistive indices and SWE are still under investigation in the research setting only [49,50].

The evaluation of other parameters, such as inhibin B levels [51] or the evaluation of the total motile sperm count (TMSC) before the intervention, would also seem to help predict the response to the treatment [52].

Further, even in the context of varicocele, the study of proteomics could play a role. The patients who showed improvement after surgery presented a greater expression of antioxidant proteins before surgery than those without improvement. Conversely, men demonstrating no improvement have expressed more proteins with a pro-inflammatory action [53].

Moreover, using AI algorithms could help combine all the parameters that can be evaluated in the pre-intervention diagnostic work-up of varicocele patients, allowing a more precise selection of the patients to undergo surgery [54]. As in other fields, the use of AI has also been proposed to improve the predictive performance of patients’ response to treatment and, therefore, the selection of the best patients on which to operate. For example, Ory and colleagues, through a random forest algorithm, highlighted how pre-repair FSH values, sperm concentration, and bilaterality were predictive of improvement [55].

Executing well-designed studies could confirm the usefulness of these parameters in the evaluation and prognosis of varicocele patients in the future.

## 3. Advances in Male Infertility Treatment

### 3.1. The Role of Oxidative Stress and Antioxidants

One important aspect of male infertility is oxidative stress (OS), which is defined as an imbalance between ROS and antioxidants (AO). OS has been linked to aberrant sperm; however, it has long been unclear exactly how this affects fertilization and conception [56]. Numerous potential mechanisms exist via which OS might adversely impact fertility, such as the disruption of sperm capacitation and sperm membrane and DNA damage, which could reduce the sperm’s ability to fertilize an egg and grow into a viable embryo [57].

A recent descriptor for males with infertility with abnormal semen parameters and OS has been proposed: Male Oxidative Stress Infertility (MOSI). This includes many patients who were previously categorized as having idiopathic male infertility. The measurement of the ORP, which considers the levels of both oxidants and reductants (antioxidants), can be a helpful clinical indicator for the classification of MOSI [57]. The assays MiOXSYS^®^ and OxiSperm^®^ II have been developed as diagnostic tools to assess OS through an accurate, fast, and repeatable measurement of the ORP.

Because AO are widely available and reasonably priced in many regions of the world, they are frequently utilized to treat male infertility [58]. Nevertheless, the current OS therapy regimens, which include the use of AO, lack a solid scientific backing, increase the risk of complications, and raise the cost of healthcare [57]. The “antioxidant paradox” is still one of the most perplexing andrological mysteries that requires close attention to detail and in-depth research [56].

With the extensive empirical usage of such supplements, the idea of “moderation” in dealing with OS and AO therapy has been developed to prevent antioxidant paradox. In light of this, developing new evidence-based treatment guidelines and a dynamic workup algorithm should be adopted. Separating males with idiopathic or unexplained infertility into MOSI-positive and MOSI-negative groups will enhance screening uniformity and help identify potentially modifiable factors [56].

The role of using AO prior to surgical sperm retrieval is further discussed later in this article.

### 3.2. The Evolution of the Indications of Varicocele Repair in Subfertile Men

According to the most recent guidelines from the European Association of Urology (EAU) [59], the American Urologic Association, and the American Society of Reproductive Medicine (AUA/ASRM) [60], a man who has a palpable varicocele, infertility, one or more abnormal semen parameters, and is in a relationship with a female who does not have fertility issues should have the varicocele repaired [59,60]. New evidence suggests varicoceles may produce pan testicular dysfunction, including both reproductive and endocrinological forms. This information has spurred varicocele repair research for other purposes [61].

#### 3.2.1. Sperm DNA Fragmentation

Varicocele is known to have a deleterious effect on sperm DNA [62,63]. A prospective study by Nguyen et al. compared 179 patients with infertility with clinical varicocele and 179 healthy volunteers who had no fertility issues and found that the sperm DNA fragmentation index (DFI) was higher in patients with varicocele, regardless of the grade, although most patients with grades II and III had DFI > 30% [64]. Additionally, varicocele repair (VR) has been reported to improve the SDF [62]. In a study, the DFI decreased from 21.6 to 11.8% following bilateral VR and from 23.0 to 12.1 following unilateral VR one year after surgery [65]. Similarly, Zaazaa et al. reported a decrease in the DFI from 34.6% to 28.3% following subinguinal VR [66]. According to a meta-analysis by Wang et al., the DFI improved by an average of 3.37% after VR [63]. Another review of 12 studies with 511 individuals showed that VR reduced the degree of sperm DNA damage [62]. 

Notably, the data indicate that higher-grade varicoceles can be linked to more DNA damage, even though additional research is required to elucidate the effect of the varicocele grade on DNA fragmentation [67,68]. Despite available evidence suggesting that DNA fragmentation could be improved with the repair of all varicocele grades, grade 3 varicocele repair leads to a larger improvement in the DFI [66]. Although insufficient evidence exists on the impact of DNA fragmentation improvements after VR on conception rates, available data indicate that couples who achieve pregnancy after VR experience less DNA damage [69,70].

#### 3.2.2. Hypogonadism

In 1975, a small case series of hypogonadal men mentioned improving serum testosterone (T) levels after VR [71]. Since then, multiple studies have demonstrated the effect of varicocele on the function of Leydig cells [72,73,74]. Furthermore, testosterone levels in men with varicoceles were significantly lower than in controls, according to other studies [75,76]. Also, varicocelectomy significantly improved serum T in hypogonadal men but not eugonadal men. T alterations were closely linked with baseline T and sperm count [75].

VR improved T levels by 97.5 ng/dL in 814 men, according to a 2012 meta-analysis [77]. The heterogeneity of the research was criticized in that analysis [61]. Chen et al. [78] conducted a meta-analysis with stricter inclusion criteria. They evaluated eight studies with 712 individuals and found a 34.3 ng/dL testosterone improvement in men diagnosed as subfertile who underwent VR. Varicocelectomy resulted in mean improvements of 123 ng/dL (*p* < 0.00001) in hypogonadal males: a greater improvement than observed in eugonadal men or the untreated group [78]. These results are consistent with research that suggests males with low or low–normal testosterone levels may benefit most from varicocelectomy in terms of testosterone increases [79].

#### 3.2.3. Non-Obstructive Azoospermia

The role of VR has been explored in the literature; this topic will be discussed later in this review.

#### 3.2.4. Orchalgia

Testicular pain related to varicocele is uncommon, and it is yet unclear if surgical correction helps to alleviate symptoms. It is anticipated that 2–10% of individuals with subclinical or clinical varicocele may experience this kind of testicular pain [80]. 

Urologists routinely perform VR to improve testicular pain [81]. However, some urologists remain reluctant to perform VR when orchalgia is the only symptom due to the absence of strong evidence of pain improvement after surgery and the still unclear mechanism by which varicocele induces testicular pain [82]. 

Other causes of testicular pain should be excluded first before proceeding with VR; these include epididymo-orchitis, testicular torsion, trauma, hydrocele, inguinal hernia, lower urinary tract infections, and referred pain [83].

A meta-analysis of 12 studies was conducted by Han et al. [84] to study the effectiveness of different varicocelectomy approaches and techniques to improve the testicular pain related to varicocele and factors that predict surgical outcomes. They found no significant correlation between the varicocele grade and pain resolution. Dull pain improved significantly more than sharp pain. The subinguinal technique resulted in significantly more pain improvement than the inguinal approach. Also, the microsurgical technique was significantly better than laparoscopic surgery at alleviating pain [84]. In contrast, Maghraby et al. found that laparoscopic varicocele treatment relieves pain [85]. External spermatic vein ligation was found to be essential for the pain alleviation that laparoscopic methods could not provide [86].

Without randomized trials comparing varicocele treatment to non-treatment or conservative treatment, definite recommendations cannot be made.

### 3.3. The Stepwise Approach to Surgical Sperm Retrieval

Approximately 1% of males globally suffer from NOA [87]. Conventional testicular sperm extraction (TESE) can be used to harvest spermatozoa blindly; however, this method is only effective in a small percentage of NOA patients [88]. First described by Schlegel and Li in 1998 [89], microsurgical testicular sperm extraction (mTESE) is now often performed [90] and has been suggested as a clinically helpful salvage treatment option for men whose TESE has failed in the past [91,92]. Later, Franco et al. described a detailed novel stepwise approach for mTESE in 2016 [93].

A wide range of expertise is needed to properly assess prospective patients and efficiently perform the mTESE process. This includes the knowledge of presentation, medical history, physical examination, and the choice and interpretation of tests, such as testicular histology, ultrasonography, hormonal and genetic testing, and semen analysis [91,94,95]. Patient selection can be optimized with the correct information and abilities [95]. 

Additional specialized expertise related to patients and the intraoperative setting helps improve mTESE success. Age, testicular volume FSH, LH, and testicular histology were helpful predictors of salvage mTESE in NOA, according to a recent systematic review and meta-analysis and other studies [96,97,98]. Moreover, some studies have suggested that noncoding RNAs and molecular biomarkers in seminal plasma could be used to predict the recovery of sperm [87]. 

### 3.4. Current Scientific Evidence Regarding The Optimization of a Patient before Surgical Sperm Retrieval and Its Impact on Sperm Yield

Given the high expenses of the mTESE strategy in the current global era of cost containment, identifying individuals who stand to gain the most from it is essential [94]. To improve the probability of sperm retrieval in men with NOA, advanced surgical techniques [99,100] and diagnostic methods, hormonal therapy, and varicocelectomy have all been proposed.

#### 3.4.1. Hormonal Therapy in NOA Prior to mTESE

In an attempt to optimize the hormonal milieu and improve the sperm retrieval rates (SRR), hormonal therapy may be prescribed before mTESE [101]. In a retrospective study, Peng et al. [102] studied the use of preoperative hCG or combined hCG and highly purified urine FSH (uFSH) versus no treatment. They found a statistically significant difference between the preoperative gonadotropin therapy and non-gonadotropin treatment groups in the SRR (31.2% vs. 19.5%, respectively, *p* = 0.006). No differences in the clinical pregnancy rate, live birth delivery rate, or miscarriage rate were observed between the two groups [102].

On the other hand, Reifsnyder et al. found in 2012 that medical therapy before mTESE was not advantageous [103].

Regarding previously failed sperm retrieval after mTESE, Shiraishi et al. [104] in 2012 studied the use of hormonal treatment prior to repeating mTESE, versus no treatment. The SRR at the second mTESE was 21% from men who had received hormonal therapy, whereas no sperm were retrieved from untreated men (*p* < 0.05). Success at the second micro-TESE was more likely if histology at the first mTESE showed hypospermatogenesis [104].

For men with Klinefelter’s syndrome (KF) and NOA, Ramasamy et al. [105] performed a retrospective analysis of data from 91 mTESE attempts that were performed on 68 men with non-mosaic KF. Men with serum T levels < 300 ng/dL received adjuvant therapy with aromatase inhibitors, clomiphene citrate, or hCG before surgery. Testicular sperm was retrieved from 45 men (66%), representing 62 attempts (68%). Men who required medical therapy and responded to treatment with serum T of 250 ng/dl or above had a greater sperm retrieval rate (77% vs. 55%). The rates of clinical pregnancy and live births for sperm-retrieved in vitro fertilization (IVF) attempts were 57% and 45%, respectively [105].

Although there is a physiological basis for adjuvant hormone therapy before testicular sperm extraction, the evidence that is now available is minimal, and there is no consensus regarding the type, dose, or duration of medication that is best for men with NOA [101].

#### 3.4.2. Antioxidants Prior to mTESE

Despite the availability of a plethora of data investigating the impact of oxidative stress and inadequate antioxidant capacity in the seminal plasma of men with infertility, most of the work currently published focuses on men with oligo/asthenozoospermia, rather than azoospermia [106].

Among the few trials on azoospermic males using antioxidants, a study was conducted by Singh et al. [107] in 2010; they included 35 azoospermic patients. A fine-needle aspiration cytology was done on all patients, showing maturation arrest, mainly at the spermatid level. These patients were divided into two groups. The untreated group (n = 11) did not receive any treatment, and the treated group (n = 24) received multivitamins, micronutrients, and co-enzyme Q10 for three months. The patients were instructed to follow up every month (n = 9). In the untreated group, the semen analysis remained unchanged in the first follow-up visit. Consequently, they failed to show up for the next follow-up visits. In the treated group, a reduction in liquefaction time and the relative viscosity of the semen was observed. Surprisingly, the authors reported that in the treated group, there was the appearance of spermatozoa (4.0 million/mL) with progressive motility (7%) and normal morphology (6%), even in the first follow-up visit (n = 6). The sperm count, motility and normal morphology increased significantly on subsequent visits. The count increased to 8 million/mL, progressive motility to 12%, and normal morphology to 9% at the second visit (n = 9). At the third visit, the count increased to 9.7 million/mL, progressive motility to 12%, and normal morphology to 13%, with a *p* value < 0.05. Within three months, two pregnancies were reported [107]. The finding of this study should be very cautiously considered as the study was poorly conducted, with the poor documentation of methodology [106], as well as the dropout of the whole control group after 1 month. Furthermore, only nine patients completed the follow-up visits, which renders the statistical results weak. Notably, the surprising results are not replicated or supported in further studies.

#### 3.4.3. Varicocele Repair Prior to mTESE

Although the deleterious effect of varicocele on semen parameters has been well established, its impact in men with NOA on the SRR and testicular histopathological patterns is yet to be explored [108]. Nevertheless, if there is a concomitant clinically palpable varicocele and NOA, clinicians may choose to perform VR prior to mTESE [101]. The most extensive systemic review and meta-analysis (SRMA) assessing VR in men with NOA was conducted by Esteves et al. [109] in 2016. According to their findings, VR yielded improved SRR compared to the control. Furthermore, about 44% of the men who underwent varicocelectomy had enough sperm in their ejaculate. Nevertheless, the existence of few data regarding the results of pregnancy outcomes using testicular or postoperative ejaculated sperm hinders the drawing of any clear conclusions about the potential for enhanced fertility in men undergoing treatment [109]. This meta-analysis has been criticized by Flannigan and Schlegel [101] for having a selection bias, as they did not include an extensive study by Schlegel and Kaufman. In their retrospective study, Schlegel and Kaufman found that only 9.6% of men who underwent VR had their sperm count recover in the ejaculate sufficiently to avoid mTESE. They also noted that the SRR were the same for men with clinical varicoceles who did not undergo VR [110].

Moreover, another SRMA by Kirby et al. [111] evaluated the role of VR in men with oligozoospermia and azoospermia before ART. Two studies on azoospermic men were identified for inclusion. Their findings showed that VR increased the likelihood of both pregnancy (OR 2.336, *p* = 0.044) and sperm retrieval (OR 2.509, *p* = 0.0001) [111].

According to the current available evidence, the role of VR in NOA is still unclear. Although its beneficial effect on increasing the SRR or even sperm appearance in the ejaculate has been concluded by some authors, more randomized controlled trials and SRMA are still needed so that VR could be routinely recommended for NOA men before mTESE.

#### 3.4.4. Gonadotropin Replacement Therapy for Male Hypogonadotropic Hypogonadism

Male hypogonadotropic hypogonadism is a gonadotropin deficiency due to hypothalamic or anterior pituitary dysfunction. If fertility is desired, gonadotropin replacement therapy is initiated [112]. The combination of hCG and recombinant human FSH (rhFSH) resulted in the appearance of sperm in the ejaculate in 88.6% of the patients. The frequently used dose of hCG was 5000 IU twice a week, and the starting dose of rhFSH was 150 IU twice a week, with the dose being increased depending on the presence of spermatogenesis [113]. The average time for sperm to appear in the ejaculate has been reported as 10.7 months after initiation of therapy [113]. There is no clear consensus regarding the duration of therapy to initiate spermatogenesis; however, when no sperm appear in the ejaculate after 1 or 2 years of gonadotropin therapy, mTESE could be done [112].

## 4. Key Points

The latest WHO Manual has made it possible to overcome many limitations of the previous edition.Although the WHO Manual has now included SDF as an extended test and the literature supports the diagnostic usefulness of SDF assessment, further efforts are needed to standardize the testing methods and establish clear reference values.Artificial intelligence will play an essential role in the future to improve the evaluation and diagnosis of male infertility.At-home semen analysis tests can be helpful in the early identification of patients with altered sperm parameters who need further investigation and wish to avoid the embarrassment some patients feel about undergoing sperm examination in a laboratory.“Omics” studies represent a significant area of interest in fertility assessment and, in the future, may make it possible to reduce the proportion of patients diagnosed with idiopathic infertility significantly.Varicocele still represents a topic of considerable clinical interest, and identifying new diagnostic indices will enable an increasingly better selection of patients for treatment.A growing body of evidence suggests that varicocele can affect the pan testicular function.Even though most men with varicocele have normal conventional semen parameters, varicocele could still affect men’s fertility, probably through increasing sperm DNA damage and decreasing intratesticular testosterone production, which indirectly affects spermatogenesis.Regardless of fertility, hypogonadism is an emerging indication for VR. Several studies reported the improvement of testosterone levels in hypogonadal men but not in euogonadal men.mTESE remains the gold standard of management for NOA. Adopting a stepwise technique and quality control measures during surgery increases the chances of sperm retrieval while minimizing trauma.Considerable research has been carried out to optimize the patients before TESE and to increase the chances for sperm retrieval using an adjuvant therapy or VR. However, no definite recommendations could be made from an evidence-based point of view.

## 5. Experts Comment

Since infertility is a global health problem with variable management approaches, it is not surprising how remarkable scientific advances are affecting this field from year to year. The diagnostics of male infertility are moving toward an increasingly accurate assessment of molecular and genetic aspects, although further studies are needed for the precise biomarkers of fertility to be identified. Once validated, these markers could become commonly used in the clinical setting of the infertility patient’s diagnostic workflow. Furthermore, the identification of new genes related to infertility will allow researchers in the future to create personalized genetic panels based on the patient’s sperm alterations, thus minimizing the proportion of patients with idiopathic infertility. Finally, progress in diagnostics, together with a conscious use of artificial intelligence, will significantly improve the management of pathologies closely associated with infertility, such as varicocele, and also improve ART outcomes. In addition, a key role in the management of male infertility in the future will be improving access to care for all men with infertility. In fact, it is well known that even in highly developed continents such as the USA, there are geographic barriers to the access to treatment for male infertility, with a distribution of urologists specialized in infertility management that is not uniform within the country [114]. This results in limited access for some patients to treatment [114]. Extending this to developing countries such as in Africa and East Asia, which have the greatest burden of male infertility today [115], one realizes the importance of breaking down geographic, socioeconomic, and financial barriers to the adequate management of male infertility [114]. In addition, the lack of awareness by the general public about the role of the urologist and reproductive specialists in infertility management often results in the underdiagnosis or misdiagnosis of male infertility [116]. In this regard, the role of specialty societies, community organizations, and the media in improving the awareness of the importance of male infertility as much as possible is crucial [114].

Despite guidelines recommending VR primarily for male infertility and abnormal semen parameters, there is a growing trend toward expanding its indications. However, more research and randomized controlled trials are needed before offering VR to patients with only hypogonadal symptoms or testicular pain, after ruling out other causes. The role of VR in non-obstructive azoospermia (NOA) remains controversial, regarding its effect on surgical sperm retrieval or sperm appearance in the ejaculate. While attempts have been made to optimize NOA patients before surgical sperm retrieval, routine adjuvant medical therapy before testicular sperm extraction (TESE) cannot be recommended until stronger evidence is available.

## Figures and Tables

**Figure 1 ijerph-21-01222-f001:**
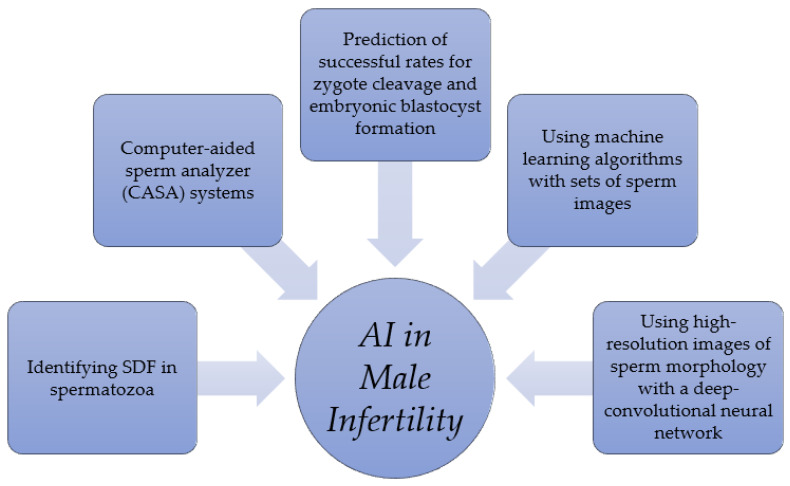
The main fields of the application of artificial intelligence in the management of male infertility to date.

**Table 1 ijerph-21-01222-t001:** Summary of the advantages and limitations of the sixth edition of the WHO Manual of Semen Analysis.

The Sixth Edition of the WHO Manual of Semen Analysis	
Advantages	Limitations
Analyzes a larger and more geographically diverse population. Specifies that the 5th centile values are only one way to interpret the results of SA, and the use of the 5th centile alone is not sufficient to diagnose male infertility.	
Second-level investigations were included in specific categories of patients: - Sperm DNA fragmentation. - Fluorescent in situ hybridization **for** genetic anomalies.	Reintroduction of the distinction between slowly progressive and rapidly progressive motility without any recent studies and which is difficult to quantify accurately by the manual method
	The absence of clear reference limits for clinicians not experienced in the field.

**Table 2 ijerph-21-01222-t002:** Summarizes the characteristics of the main home semen analysis kits [25].

Type of Kit	Type of Outcome Assessed	Advantages	Limitations
SpermCheck Fertility	Sperm concentration above or below 20 mil	Rapid (10 min) Simple to interpret	No information about the exact value of sperm concentration. No information about other parameters.
Micra Sperm Test	Sperm count, sperm motility, and sperm volume	Rapid (30 min) It allows the quantification of three different parameters	Risk of interpretation error by the observer. No evaluation of morphology
Trak	Sperm count	Rapid (6 min) Simple to interpret	No information about the exact sperm concentration. No information about other parameters.
Fertell Male Fertility Home Test	Sperm motile count	Rapid Simple to interpret	No information about the exact value of sperm motile count. No information about other parameters.
SwimCount Sperm Quality Test	Progressive sperm motile count	Rapid (1 h) Simple to interpret	No information about the exact value of progressive sperm motile count. No information about other parameters.

## Data Availability

No new data were created or analyzed in this study. Data sharing does not apply to this research.

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
