# Peer review of "The Andrological Landscape in the Twenty-First Century: Making Sense of the Advances in Male Infertility Management for the Busy Clinicians"

_ijerph, 2024, doi:10.3390/ijerph21091222_

Round 1
Reviewer 1 Report
Comments and Suggestions for Authors
This review, entitled “Andrological Landscape in the Twenty-First Century: Making Sense of the advances in Male Infertility for the Busy Clinicians” addresses various aspects of the medical approach to diagnosing male infertility and critiques the recommendation of the SDF and FISH tests in the current WHO manual Semen analysis. It also analyzes the limitations of these tests and provides physicians with recommendations for their use. This includes supporting means that use artificial intelligence to improve the analytical process such as CASA, and the benefits and limitations of home tests to determine sperm fertility in semen. This review also examines the contribution of omic sciences to the diagnosis of infertility and addresses the possible genetic identification of alterations, particularly in patients with idiopathic infertility. Addresses the problem of diagnosing varicocele as a cause of infertility and proceeds to treats this alteration for treatment of male infertility, covering topics such as sperm DNA fragmentation, hypogonadism, orchialgia, surgical sperm retrieval, antioxidant, hormone therapy and correction of varicocele prior to TESE.
The present review gives us a very comprehensive overview of the problem of male infertility, as we believe that its publication is relevant in addition to the authors’ expert opinion on these issues. Therefore, I consider that this review can be published in your journal, with the only remark being to delete point 3.13, which is not addressed in this part of the review, but only in 3.2.
Author Response
Comment 1:
This review, entitled “Andrological Landscape in the Twenty-First Century: Making Sense of the advances in Male Infertility for the Busy Clinicians” addresses various aspects of the medical approach to diagnosing male infertility and critiques the recommendation of the SDF and FISH tests in the current WHO manual Semen analysis. It also analyzes the limitations of these tests and provides physicians with recommendations for their use. This includes supporting means that use artificial intelligence to improve the analytical process such as CASA, and the benefits and limitations of home tests to determine sperm fertility in semen. This review also examines the contribution of omic sciences to the diagnosis of infertility and addresses the possible genetic identification of alterations, particularly in patients with idiopathic infertility. Addresses the problem of diagnosing varicocele as a cause of infertility and proceeds to treats this alteration for treatment of male infertility, covering topics such as sperm DNA fragmentation, hypogonadism, orchialgia, surgical sperm retrieval, antioxidant, hormone therapy and correction of varicocele prior to TESE.
Response 1:
Thank you for your valuable feedback and encouraging words. We are glad that our work has met your expectations.
Comment 2:
The present review gives us a very comprehensive overview of the problem of male infertility, as we believe that its publication is relevant in addition to the authors’ expert opinion on these issues. Therefore, I consider that this review can be published in your journal, with the only remark being to delete point 3.13, which is not addressed in this part of the review, but only in 3.2.
Response 2:
Point 3.1 discussed “Evolution of indications of Varicocele repair in subfertile men”, since NOA is one of the emerging indications for VR, we added it as point 3.1.3 in order not to be missed in this context. However, to avoid repetition of the information we mentioned the details once in point 3.3.3.
Reviewer 2 Report
Comments and Suggestions for Authors
GENERAL COMMENTS
I read with much enthusiasm the article “Andrological Landscape in the Twenty-First Century: Making Sense of the Advances in Male Infertility for the Busy Clinicians,” which was recently submitted to the International Journal of Environmental Research and Public Health
The authors are to be commended for providing useful up-to-date knowledge for the latest advances in male infertility. It is indeed a harsh reality that infertility is a common and significant burden for male health.
Overall, the manuscript is well-written, detailed, clear, and concise. The authors review the relevant literature pertinent to the latest diagnostic and therapeutic pathways of male infertility offering a clear-cut overview on the subject. They carefully convey the message that several issues in andrology demand answers and need to be resolved and validated in future trials. Also, they conclude with several key points and expert comments on the field.
In spite of being aware beforehand that an exhaustive analysis of the literature is not easy to conduct, I still advise the authors to make the following significant amendments:
1st Comment
Although engaging, the abstract should be reader-friendly and typed as a single paragraph with no paragraph indentation.
Also, the following remarks should be added at the end of the discussion section before the key points section.
2nd Comment
Nowadays, reproductive urologists may face significant barriers in providing optimal care, which constitutes a challenge in the 21st century. Limited availability of technical means, awareness among healthcare professionals, and financial constraints all limit urologists’ involvement. Engaging urologists more effectively in infertility management is key to optimizing fertility outcomes among couples. Add the following article PMID: 37893553
3rd Comment
A significant threat that is probably on the horizon is that global demographics shift toward increasing paternal age. This poses an additional challenge to the andrologist and might necessitate sperm cryopreservation, lifestyle modifications, and preimplantation genetic testing to ensure and optimize the best possible ART outcomes.
Informing infertile couples of the alarming correlations between older fathers and a rise in their offspring's diseases is crucial so that they can be effectively guided through their reproductive years, and this is another point to be discussed.
Add the following articles PMID: 38792276 & PMID 36833413
4th Comment
The authors unintentionally omitted to mention a very interesting point that will be exhaustively discussed in the future andrological landscape. Although they refer to OS in the pre-mTESE setting, they need to discuss OS as a critical surrogate marker that needs to be evaluated and regulated in patients with impaired semen parameters (several measurement techniques are currently available, both direct and indirect methods). They should comment on the importance of maintaining the so-called Redox balance. Recently, it was advocated that redox homeostasis is paramount in protecting fertility potential.
Add the following articles PMID: 34679669
5th Comment
Last but not least, the issue of antioxidant administration is not analyzed inadvertently. When used cautiously, antioxidants have the potential to enhance sperm quality and alleviate male infertility. Otherwise, there are risks from the over-the-counter supplementation. The latter also poses a threat. Hence, the paper can benefit from including more information derived from the following articles.
Add the following articles PMID: 32294030 & PMID: 33563149 & PMID: 34356300
Author Response
GENERAL COMMENTS
I read with much enthusiasm the article “Andrological Landscape in the Twenty-First Century: Making Sense of the Advances in Male Infertility for the Busy Clinicians,” which was recently submitted to the International Journal of Environmental Research and Public Health
The authors are to be commended for providing useful up-to-date knowledge for the latest advances in male infertility. It is indeed a harsh reality that infertility is a common and significant burden for male health.
Overall, the manuscript is well-written, detailed, clear, and concise. The authors review the relevant literature pertinent to the latest diagnostic and therapeutic pathways of male infertility offering a clear-cut overview on the subject. They carefully convey the message that several issues in andrology demand answers and need to be resolved and validated in future trials. Also, they conclude with several key points and expert comments on the field.
In spite of being aware beforehand that an exhaustive analysis of the literature is not easy to conduct, I still advise the authors to make the following significant amendments:
Response:
Thank you so much for your constructive feedback and encouraging words.
1st Comment
Although engaging, the abstract should be reader-friendly and typed as a single paragraph with no paragraph indentation.
Also, the following remarks should be added at the end of the discussion section before the key points section.
Response 1:
The manuscript, including the abstract, was formatted according to the journal guidelines for the authors. So, we apologize for not being able to change the formatting or the style of writing.
2nd Comment
Nowadays, reproductive urologists may face significant barriers in providing optimal care, which constitutes a challenge in the 21st century. Limited availability of technical means, awareness among healthcare professionals, and financial constraints all limit urologists’ involvement. Engaging urologists more effectively in infertility management is key to optimizing fertility outcomes among couples. Add the following article PMID: 37893553.
Response 2:
We thank the reviewer for the valuable comment and agree with him/her that one of the limitations in the adequate management of male infertility is the geographic, socioeconomic, and financial gaps between the different countries. Furthermore, we also know how the correct management of male infertility is influenced by the low awareness of the general population about the impact of the disease and the role of the urologist in its management. We have tried to express this thought in the section entitled Experts Comment (highlighted).
3rd Comment
A significant threat that is probably on the horizon is that global demographics shift toward increasing paternal age. This poses an additional challenge to the andrologist and might necessitate sperm cryopreservation, lifestyle modifications, and preimplantation genetic testing to ensure and optimize the best possible ART outcomes.
Informing infertile couples of the alarming correlations between older fathers and a rise in their offspring's diseases is crucial so that they can be effectively guided through their reproductive years, and this is another point to be discussed.
Add the following articles PMID: 38792276 & PMID 36833413
Response 3:
We totally agree that advanced paternal age is an important factor that might affect the reproductive and ART outcomes and the offsprings’ wellbeing. However, we were constrained with the pre-approved outline of the manuscript by the editor including the main titles and subtitles. Thus, we will seriously consider addressing this important issue in a future publication.
4th Comment
The authors unintentionally omitted to mention a very interesting point that will be exhaustively discussed in the future andrological landscape. Although they refer to OS in the pre-mTESE setting, they need to discuss OS as a critical surrogate marker that needs to be evaluated and regulated in patients with impaired semen parameters (several measurement techniques are currently available, both direct and indirect methods). They should comment on the importance of maintaining the so-called Redox balance. Recently, it was advocated that redox homeostasis is paramount in protecting fertility potential.
Add the following articles PMID: 34679669
5th Comment
Last but not least, the issue of antioxidant administration is not analyzed inadvertently. When used cautiously, antioxidants have the potential to enhance sperm quality and alleviate male infertility. Otherwise, there are risks from the over-the-counter supplementation. The latter also poses a threat. Hence, the paper can benefit from including more information derived from the following articles.
Add the following articles PMID: 32294030 & PMID: 33563149 & PMID: 34356300
Response to the 4th and 5th comment:
Thank you for pointing this out. The matter of OS and antioxidants is of vast importance. We have added a paragraph discussing this issue (highlighted in the main manuscript).
Reviewer 3 Report
Comments and Suggestions for Authors
The authors did very good job in summarizing the basic concepts and advances in male infertility, however, there is zero graphs and tables in this manuscript. I strongly urge the authors should add more graphs and tables, helping the readers more efficiently read this article, and hence increase the overall quality of this article.
Author Response
Reviewer 3
The authors did very good job in summarizing the basic concepts and advances in male infertility, however, there is zero graphs and tables in this manuscript. I strongly urge the authors should add more graphs and tables, helping the readers more efficiently read this article, and hence increase the overall quality of this article.
Response:
Thanks a lot for your valuable feedback and important suggestions.
Table 1 was already included which summarizes the characteristics of the main home semen analysis kits.
We also added another table summarizing the advantages and limitations of the 6th Edition of the WHO manual of Semen Analysis (Table 1).
We added a figure demonstrating the role of AI in male infertility.